# Successful Silencing of the Mycotoxin Synthesis Gene *TRI5* in *Fusarium culmorum* and Observation of Reduced Virulence in VIGS and SIGS Experiments

**DOI:** 10.3390/genes13030395

**Published:** 2022-02-23

**Authors:** Polina Tretiakova, Ralf Thomas Voegele, Alexander Soloviev, Tobias Immanuel Link

**Affiliations:** 1Central Research Institute of Epidemiology, Federal Service for Surveillance on Consumer Rights Protection and Human Wellbeing, Novogireevskaya Street 3A, 111123 Moscow, Russia; 2Department of Phytopathology, Institute of Phytomedicine, Faculty of Agricultural Sciences, University of Hohenheim, Otto-Sander-Straße 5, 70599 Stuttgart, Germany; ralf.voegele@uni-hohenheim.de; 3All-Russia Research Institute of Agricultural Biotechnology, Timiryazevskaya Street 42, 127550 Moscow, Russia; a.soloviev70@gmail.com

**Keywords:** *F. culmorum*, FHB, dsRNA, RNAi, VIGS, SIGS, trichothecenes, virulence, gene expression

## Abstract

Crops constantly experience various biotic stresses during their life cycle, and *Fusarium* spp. remain one of the most serious groups of pathogens affecting plants. The ability to manipulate the expression of certain microorganism genes via RNAi creates the opportunity for new-generation dsRNA-based preparations to control a large number of diseases. In this study, we applied virus-induced gene silencing (VIGS), and spray-induced gene silencing (SIGS) to silence the trichothecene-producing gene *TRI5* in *F. culmorum* as a means to reduce its aggressiveness on spring wheat. Treatment of the fungus with dsTRI5RNA in vitro reduced deoxynivalenol (DON) and 3-acetyldeoxynivalenol (3-A-DON) accumulations by 53–85% and 61–87%, respectively, and reduced *TRI5* expression by 84–97%. VIGS decreased the proportion of infected wheat spikelets by 73%, but upregulation was observed for *TRI5*. SIGS on wheat leaves and ears using certain dsTRI5RNA amounts negatively impacted *F. culmorum* growth. However, when performing in vivo analyses of *TRI5* mRNA levels, the upregulation of the gene was determined in the variants where fungal colonization was restricted, suggesting a compensatory reaction of the pathogen to RNAi.

## 1. Introduction

*Fusarium culmorum* and *Fusarium graminearum* are spread throughout the world, affecting a large number of crops. Importantly, they cause fusarium head blight (FHB), a wilting of the ears of grain species after infection through the floret. They pose a threat to animal and human health through toxin-contaminated plant material. Both species are able to produce significant amounts of trichothecene mycotoxins that cause severe food poisoning. The predominant trichothecenes produced are deoxynivalenol (DON), nivalenol (NIV), 3-acetyldeoxynivalenol (3-A-DON), and 15-acetyldeoxynivalenol (15-A-DON). The high virulence of *F. culmorum* and *F. graminearum,* combined with the regular detection of trichothecenes in contaminated plant tissue, suggests that the toxins are virulence factors in certain hosts. It has been reported that the fungi produce trichothecenes right before actual tissue colonization, weakening the host and most likely leading to an inhibition of plant immune responses [1]. Trichothecenes suppress protein biosynthesis [2]. Thus, plant defense reactions can be disrupted or delayed, as an effective immune response directly depends on transcription and translation processes [1].

This theory is supported by additional observations. *F. graminearum* mutants could not quickly colonize the area of the wheat head adjacent to the inoculated spikelet without DON secretion. Interestingly, the fungus was still able to penetrate the plant tissue at the point of inoculation [3]. Eudes et al. [4] observed that the inability of *F. graminearum* to produce trichothecenes resulted in a slower pathogen spread within plant ears, to the point of a complete restriction of growth to the infection site. Maier et al. [5] reported that even though *F. graminearum* strains with a disrupted *TRI5* gene caused FHB symptoms on inoculated wheat ears, they were not able to spread to other wheat ears. Without trichothecenes, *F. graminearum* strains could not grow quickly on wheat, triticale, rye, or barley [6]. A positive correlation was demonstrated between the amount of toxins produced and the virulence of *F. graminearum* and *F. culmorum* strains [7,8,9,10].

One way to control *Fusarium* spp. is the application of different fungicides. However, their effectiveness often depends on multiple factors that cannot be controlled. In some cases, fungicides can induce toxin accumulation, especially under conditions unfavorable for fungal growth or if fungicide concentrations are insufficient [11]. Additionally, pathogens keep evolving and may develop tolerances to fungicides, as well as host defenses. This makes the search for new methods to control FHB mandatory.

Recently, dsRNA-based preparations have appeared as a novel strategy with a high potential to control pathogens including *Fusarium* spp. It was found that dsRNA is also important in plant-pathogen “communication”. The exchange of small RNAs (sRNAs) between pathogen and host has been observed in many pathosystems and is now called RNA trafficking [12]. 

A good example of sRNA movement from one species to another is the method of host-induced gene silencing (HIGS), along with virus-induced gene silencing (VIGS) and spray-induced gene silencing (SIGS).

Over the past few years, HIGS has been effectively used to study host-pathogen interactions and gene function. HIGS is based on plant transformation, resulting in the expression of exogenous dsRNA targeting genes of interest in the pathogen. It has been used on plants affected by nematodes [13], insects [14,15], and fungi [16,17,18,19]. The VIGS method utilizes viral vectors such as barley stripe mosaic virus (BSMV), allowing transient synthesis of dsRNA without plant transformation. The BSMV-system was established and used to silence barley and wheat genes [20,21,22,23]. 

SIGS is a more recent technique which is based on in vitro dsRNA production and the spraying of dsRNA onto the plant surface. This not only avoids time-consuming plant transformation or complex manipulations with viral vectors, but is an entirely novel approach to crop protection. The technique was applied against the fungal pathogens *F. graminearum* [24,25], *Rhizoctonia solani*, *Aspergillus niger*, *Verticillium dahliae* [26], and *Podosphaera xanthii* [27], and against the oomycetes *Hyaloperonospora arabidopsidis* [28], and *Phytophthora infestans* [29].

Interestingly, sRNAs are not only able to trigger mRNA degradation via RNAi, but are also able to initiate the production of siRNA. siRNA can therefore either be primary as a result of Dicer cutting dsRNA, or secondary from biosynthesis by RNA-directed RNA polymerase [30]. Both types of siRNA are involved in RNAi [31,32,33]. This property is of high value when developing dsRNA-based strategies to control diseases. The stability of exogenous dsRNA is restricted to approximately one week on plant surfaces. However, its effectiveness is higher and longer-lasting when it first enters the plant, for example, through wounds and then is absorbed by a fungus. 

dsRNA-based preparations potentially have many advantages over conventional pesticides. Such preparations can be made specific to a particular species if it is able to perform RNAi. Ecologically this is a very valuable feature, since it can effectively avoid the killing of beneficial species. dsRNA degrades quickly and can be used in combinations targeting different genes or gene regions. This provides more options for the modification of preparations once pathogens develop resistance [24,34]. The development of dsRNA preparations has the potential to be much faster than that of chemically synthesized fungicides, which can also dramatically reduce the costs of new fungicides. The production costs for dsRNA are still much higher than for conventional fungicides, but they have reduced substantially over the last few years [35]. Additionally, the production of dsRNA is possible without chemical synthesis, which could make them applicable in organic agriculture once they are approved. Thus, RNAi can be developed into a highly environmentally friendly method for pathogen control, that is also cost effective.

Nevertheless, there are many issues yet to be resolved when using RNAi to control pathogens on a large scale, such as rapid dsRNA degradation, high production costs and off-target effects [34]. 

Non-specific gene silencing can be a serious hurdle when developing dsRNA-based preparations. This may happen when dsRNA molecules get cut by Dicer into siRNAs that become complementary to other mRNAs [36]. Thus, not only the host itself but also beneficial microbes could suffer. 

In this research, we tested the effect of dsRNA interfering with *F. culmorum* trichothecene production on fungal virulence when colonizing spring wheat. A dsRNA of 161 bp in length complementary to the *TRI5* gene was used. *TRI5* encodes trichodiene synthase, catalyzing the first step of trichothecene biosynthesis. To test the effectiveness of dsTRI5RNA against toxin accumulation, we first applied it to *F. culmorum* liquid culture. This was followed by in vivo experiments on wheat. We used VIGS and SIGS approaches to analyze *F. culmorum* virulence on wheat plants and detached leaves.

Earlier investigations have focused mostly on the silencing of “house-keeping” genes as a potential means to control *Fusarium* spp. However, we think that the manipulation of pathogen virulence factors could be an effective approach when developing new methods for crop protection. Eventually, we expect our findings to help reduce crop losses caused by trichothecene contamination.

## 2. Materials and Methods

### 2.1. Plant Material

Two varieties of spring wheat were obtained from the Russian Timiryazev State Agrarian University (Moscow, Russia). They were selected because of their different responses to FHB, observed in trial fields of the Russian Timiryazev State Agrarian University. Although both varieties are susceptible to FHB under the conditions that are favorable for the fungus, Zlata tolerates the disease significantly better than Ivolga (data not presented).

Plants were grown in a greenhouse at 20–25 °C by day and 16–20 °C by night, with a day/night regime of 16/8 hours. Plants were watered once every other day. The liquid nutrient solution WUXAL Super (AGLUKON Spezialduenger GmbH & Co. KG, Düsseldorf, Germany) was used for fertilization according to the manual. Fertilizer was applied every 10–14 days starting from the third week after wheat germination (300 mL per pot). The soil used was a 1:1 mixture of seedling substrate (Klasmann-Deilmann GmbH, Geeste, Germany) and standard soil (Gebrüder Patzer GmbH & Co. KG, Sinntal, Germany).

### 2.2. Fungal Material

*F. culmorum*, strain C99, was isolated and identified at the Russian Timiryazev State Agrarian University from samples of spring triticale (line C99). For long-term growth, SNA-agar was used [37]. For rapid growth, the fungus was cultivated on potato-dextrose broth or agar (PDB/PDA). TBI medium (trichothecene biosynthesis induction medium, pH 4.5) was used to induce trichothecene biosynthesis [38].

### 2.3. In Silico Experiments

The primers used in this study are presented in Table 1.

Primer design was carried out using Primer3 (https://primer3.ut.ee/ (accessed on 5 May 2016)). The secondary structure of primers was checked using the OligoAnalyzer™ tool (https://www.idtdna.com/pages/tools/oligoanalyzer (accessed on 5 May 2016)). Primer specificity was tested via BLAST (https://blast.ncbi.nlm.nih.gov/Blast.cgi (accessed on 5 May 2016)).

Fc-TRI5-T7 primers were tailed with the sequence of the T7-RNA polymerase promoter at the 5’-end to produce dsTRI5RNA in vitro. The resulting amplicon is located in the middle (from bases 542 to 702) of the exon sequence of the *TRI5* gene (GenBank: AY130291.1). The size of dsRNA was selected in the range of 150–250 bp to test the effectiveness of shorter molecules. Accordingly, the Fc-TRI5-REN forward and reverse primers were tailed with the restriction sites *Not*I and *Pac*I at the 5’-end to modify the BSMV vector.

The TRI5_A primers were designed for RT-qPCR to produce short (100 bp) amplicons outside of the sequence used to obtain dsTRI5RNA. This amplicon is located from bases 929 to 1028 (closer to the 3’-end) of the *TRI5* exon region. The EF1A primers produce the amplicon from bases 1433 to 1520 (88 bp) of the *EF1A* sequence (GenBank: KT008433.1).

Primer efficiency was tested by a 10-fold dilution series of *F. culmorum* DNA. The TRI5_A efficiency amounted to 90%, while the EF1A efficiency was 92%.

*TRI5* expression was calculated according to the ΔΔCt method, after a correction for efficiency using GenEx 6.0.1.612 (MultiD Analyses AB, Göteborg, Sweden).

### 2.4. Nucleic Acid Extraction

#### 2.4.1. DNA

Plasmid DNA was extracted using the PeqGOLD plasmid miniprep kit (VWR International GmbH, Darmstadt, Germany) according to the manufacturer’s manual.

*F. culmorum* was grown in PDB at room temperature with constant shaking for 3 days. Mycelia were collected, and DNA was isolated using the PeqGOLD fungal DNA mini kit (VWR International GmbH, Darmstadt, Germany) according to the manufacturer’s manual. Alternatively, DNA was extracted from fungal mycelium collected from PDA plates, in accordance with Liu et al. [41].

Fungal mycelia, as well as wheat leaves, were homogenized using a FastPrep-24™ classic instrument (MP Biomedicals GmbH, Eschwege, Germany) with liquid nitrogen and sterile steel spheres for RNA extraction. RNA isolation was performed using a plant RNA isolation mini kit (Agilent Technologies, Inc., Wilmington, NC, USA.) according to the manufacturer’s manual.

#### 2.4.2. RNA

Wheat grains were ground using liquid nitrogen, mortar, and pestle. A total of 100 mg of the obtained flour was placed in a tube with 500 µL of RNA extraction buffer (0.1 M Tris (pH 8.0), 5 mM EDTA (pH 8.0), 0.1 M NaCl, 0.5% SDS, and 1% 2- mercaptoethanol). After centrifugation, extraction of the supernatant was performed using phenol (pH 4.3)/chloroform (1:1). RNA was precipitated with isopropanol, and the pellet was washed with 70% ethanol, dried, and resolved in 100 µL of sterile RNAse-free water.

### 2.5. Vector Modification

The BSMV vectors were kindly provided by the Northwest Agriculture and Forestry University (Yangling, PRC). The γ-vector was modified to contain the restriction sites *Not*I and *Pac*I [20,42]. The targeted 161 bp sequence was amplified using Fc-TRI5-REN primers (Table 1), using *F. culmorum* DNA as a template and introducing the restriction sites *Not*I and *Pac*I. The PCR reaction mixture contained 5 µL 10× PCR-buffer, 2 µL MgCl_2_ (25 mM), 0.2 µL Taq-polymerase (1.25 units), 4 µL dNTP mixture (2.5 mM), 1 µL DNA template (0.25 µg/µL), 2 µL primer mixture (10 pmol/µL), and 36 µL of sterile autoclaved H_2_O in a total volume of 50 µL (Thermo Fisher Scientific Inc., Waltham, USA). The amplification program: 1 cycle of denaturation for 3 min at 95 °C; 40 cycles of denaturation for 30 s at 95 °C, annealing for 30 s at 59 °C, and elongation for 30 s at 72 °C, followed by final elongation for 5 min at 72 °C. The PCR product was separated on a 0.8% agarose gel, and extracted using the peqGOLD Gel Extraction Kit (VWR International GmbH, Darmstadt, Germany) according to the manufacturer’s manual. The purified amplicon and vector were treated with *Not*I and *Pac*I (New England Biolabs, Frankfurt, Germany). The amplicon was inserted into the vector using T4-DNA-ligase (Thermo Fisher Scientific Inc.).

The resulting plasmid was transformed into *Escherichia coli* DH5α using the heat-shock method. Competent cells were mixed with 1 µL of the ligation mixture and incubated for 30 min on ice. Heat shock comprised 45 s at 42 °C, followed by placing the cells on ice for 2 min. Subsequently, 300 µL SOC medium (0.5% yeast extract, 2% trypton, 10 mM NaCl, 2.5 mM KCl, 10 mM MgCl_2_, 10 mM MgSO_4_, 20 mM glucose) was added and the tubes were incubated for 1 h at 37 °C. Finally, 140 µL of the suspension was transferred to LB-plates (in 1 L: 10 g trypton, 5 g yeast extract, 5 g NaCl, 15 g agarose; pH 7.0) with 50 µg/mL of ampicillin [43]. Transformed cells were selected and transferred to new LB-plates with ampicillin using colony PCR.

### 2.6. dsRNA Transcription In Vitro

In vitro synthesis of dsRNA was done using the HiScribe T7 high yield RNA synthesis kit (New England Biolabs, Frankfurt, Germany) according to the manufacturer’s manual. DsTRI5RNA was purified in two steps. First, samples were treated with DNAseI and RNAse T1 (Thermo Fisher Scientific Inc.) to remove ssRNA and DNA molecules. Second, protein molecules were removed using phenol and chloroform. dsRNA concentration was measured using a spectrophotometer.

### 2.7. Wheat Inoculation with BSMV

Prior to viral RNA synthesis via the HiScribe T7 high yield RNA synthesis kit (New England Biolabs, Frankfurt, Germany), plasmids were linearized using the respective restriction enzymes: pα – *Mlu*I; pβ – *Spe*I; pγ – *Mlu*I; pγ-TRI5 – *Bss*HII; pγ-PDS – *Bss*HII. The entire process of RNA preparation and subsequent plant infection was performed according to Huang [42]. The process included the following steps: preparation of the viral RNAs, wheat seedling inoculation with the respective mixture of RNAs, BSMV symptom evaluation. Depending on the treatment, the following RNA mixtures were prepared: α + β + γ; α + β + γ-TRI5; α + β + γ-PDS. To confirm the effectiveness of the BSMV silencing system, a vector containing the *PDS* gene sequence was used, since the silencing of *PDS* results in the appearance of white, pigment-free areas on the leaves. The mixture of RNAs was added to a tube with FES-buffer [20], and used to inoculate the fully developed second leaf of a 10-day old wheat seedling by rub inoculation. FES-buffer contains abrasive particles that injure the leaf and enable viral RNA to enter the plant. BSMV symptoms appeared around 10 days after inoculation. The effectiveness of infection was assessed visually according to the appearance and spread of yellow stripes on a scale from 0 to 3 (0 = no symptoms, 1 = weak yellow stripes, 2 = light-yellow stripes, 3 = bright yellow stripes). In total, 108 plants were used in the BSMV experiment (36 plants in the first experiment and 72 plants in the second experiment). One plant was seeded per pot and pots were organized randomly in blocks.

### 2.8. dsRNA Application in Liquid Medium

100 µL *F. culmorum* conidia suspension (3 × 10^4^ conidia/mL) was cultivated in 2 mL of TBI medium (trichothecene biosynthesis induction medium). Different amounts of dsTRI5RNA (96 µg, 9.6 µg, 0.96 µg, and 0.096 µg, respectively) were diluted in 100 µL of sterile water and applied to the tubes 1 day later, alongside 100 µL of H_2_O as the control variant. The variants were organized in triplicates [19]. Four days later, fungal mycelia were separated from the media and used for RNA extraction, while the media was used to measure trichothecene concentrations by HPLC.

### 2.9. Trichothecene Measurement

Nivalenol (NIV), deoxynivalenol (DON), 3-acetyl-deoxynivalenol (3-A-DON), 15-acetyl-deoxynivalenol (15-A-DON), and zearalenone (ZEN) were measured in TBI medium by HPLC. TBI (1.5 mL) was transferred into 2 mL-tubes and centrifuged for 5 minutes at 12,000 rpm. One mL of the supernatant was transferred into a new tube with 0.3 g NaCl. Methyl-acetate was added to the tube. The phase with methyl-acetate was transferred into a new tube twice. Then, the suspension was concentrated at 60 °C. One mL of acetonitrile:water (1:9) was added to the tubes and centrifuged at 12,000 rpm for 5 min. Then, the mixture was transferred into new tubes and the concentration of mycotoxins was measured.

HPLC was run on an Accela™ system (Thermo Fisher Scientific Inc., Waltham, MA, USA) with LTQ velos (Thermo Fisher Scientific Inc., Waltham, MA, USA) as mass spectrometry identifier. The column used was a YMC-Trias C18 (YMC Europe GmbH, Dinslaken, Germany), an organic/inorganic hybrid silica column with particle sizes 1.9, 3, 5 µm and a pore size of 12 nm. As eluates water/acetonitrile 95/5 (A) and acetonitrile (C) were used with a gradient ranging from 10% C to 100% C over eleven minutes, and back to 10% C after that, running at a flow rate of 0.5 mL/min. 6 µL were injected. The standards employed were NIV, DON, 3-A-DON, 15-A-DON, and ZEN. The toxin concentration was calculated using formula (1), where C1 represents the toxin concentration in µg/mg dry matter (DM), C2 represents the toxin concentration in µg/mL, 2 represents the conversion to 2 mL of TBI medium, and m is the mass of dried mycelium in mg.
C1 = C2 × 2 /m(1)

### 2.10. Spraying Plants with dsRNA

The detached leaves of 14-day old wheat seedlings to be sprayed with dsTRI5RNA were transferred to plates with 1% agarose. For sterilization, leaves were submerged in 1% sodium hypochlorite solution for 30 s followed by washing with sterile water for 1 minute. Depending on the treatment (10 µg or 15, 1.5, 0.15, and 0.015 µg of dsRNA, respectively), dsTRI5RNA was diluted in 100 µL of sterile water and the whole volume was sprayed evenly onto leaves using sterile 5 mL-spraying bottles. Additionally, 15 µg/100 µL of dsRNA synthesized on the template of plasmid DNA carrying the T7-promoter sequence was used as a control for the non-specific effect of the double-stranded molecule. After spraying, leaves were left to dry for 15 minutes and then point-inoculated with a droplet (5 µL) of *F. culmorum* conidia (2 × 10^4^ conidia/mL). Plates were incubated on the lab bench for 5 days at room temperature [25]. After visual assessment, the leaves were collected for RNA extraction, further fungal biomass quantification, and evaluation of *TRI5* expression.

Each wheat head was sprayed with 500 µL of dsTRI5RNA solution depending on the treatment (15, 1.5, 0.15, and 0.015 µg of dsTRI5RNA diluted in sterile water). As in the experiment using leaves, plasmid dsRNA (15 µg/500 µL) was used as the control for non-specific silencing. In total, 36 plants were used in the experiment. Heads were left to dry for 2–3 h and subsequently point-inoculated with 10 µL of fungal conidia (2 × 10^4^ conidia/mL). A randomized complete block design was used.

### 2.11. Plant Inoculation with F. culmorum

The concentration of *F. culmorum* conidia was assessed using a hemocytometer. Fungal mycelia were collected from PDA and added to a tube with sterile autoclaved water. Mycelia were removed by filtration, and conidia were centrifuged and subsequently suspended in 1 mL of sterile autoclaved water. Wheat ears were point-inoculated with *F. culmorum* conidia at flowering time [44]. Heads were wrapped in a plastic bag lightly sprayed with water for 3 days to maintain a high humidity. When inoculating with modified BSMV, the proportion of infected florets was checked visually 7, 12, 16, and 20 days after inoculation. When spraying with dsTRI5RNA, the FHB evaluation was conducted at 7, 14, and 20 days post infection. Two weeks after *F. culmorum* inoculation, heads were collected from half of the plants for qPCR and gene expression analyses. The rest of the plants were left and monitored visually.

### 2.12. Real-Time PCR

cDNA synthesis was conducted using the RevertAid first strand cDNA synthesis kit (Thermo Fisher Scientific Inc.) according to the manufacturer’s manual. Except for in experiments with BSMV, oligo(dT)12–18 primers (Thermo Fisher Scientific Inc.) were used. Since oligo(dT) primers hybridize to the poly-A-tail and viral RNAs lack the poly-A-sequence, random hexamer primers (Thermo Fisher Scientific Inc.) were used to produce cDNA from RNA samples containing BSMV. Real-time PCR was conducted using the CFX96 Touch™ (Bio-Rad Laboratories, Inc., Hercules, CA, USA) and 96-well PCR plates. For amplification, SensiFAST™ SYBR® No-ROX kit (Bioline, London, UK) was used. The real-time PCR program: denaturation for 5 min at 95 °C; 40 cycles of denaturation for 5 s at 95 °C, annealing for 15 s at 55 °C; melt curve 65 to 95 °C for 5 s, increment 0.5 °C. *TRI5* was normalized to the *F. culmorum* translation elongation factor-1 alpha (*EF1A*). 

To determine differences in the growth of *F. culmorum* using real-time PCR, we used relative quantification and calculated a simple ratio, R. The idea was to normalize fungal RNA against wheat RNA using primers specific for each organism. When choosing stably expressed reference genes, it can be assumed that the amount of nucleic acid is directly proportional to the amount of the respective organism. A primer pair specific for the fungus amplified the region of the gene of the translation elongation factor 1 alpha 1 (*EF1A*), for wheat the actin gene (*Act*). The stability and suitability of these genes for various analyses have been confirmed in other experiments [39,45]. We used the formula:R = 2^Ct(*Act*) – Ct(*EF1A*)^(2)
where R is the ratio describing the amount of *F. culmorum*, Ct(*Act*) is the threshold cycle for *Act*, and Ct(*EF1A*) that for *EF1A*.

### 2.13. Statistical Analyses

Data analyses were performed using R-studio (http://www.rstudio.com/ (accessed on 3 August 2019)). Statistical evaluation was conducted using one-way ANOVA and the LSD test.

## 3. Results

### 3.1. The effect of dsTRI5RNA on Trichotecene Biosynthesis in F. culmorum Liquid Culture

The aim of the in vitro experiments was to test the effectiveness of dsTRI5RNA application in the silencing of the *TRI5* gene and, therefore, on deoxynivalenol (DON) and 3-acetyl-deoxynivalenol (3-A-DON) production. To synthesize dsRNA, a 161 bp gene fragment was chosen (see Section 2.3) that has no homology with the sequences of other organisms present in public databases, particularly with the wheat genome, thus excluding the non-specific silencing of plant genes.

The long-term cultivation of fungi on artificial media often results in an inability to produce toxins. To ensure the synthesis of trichothecenes, TBI medium was used (see Section 2.2).

HPLC showed that the *F. culmorum* strain C99 was able to produce two types of trichothecenes – DON and its derivative 3-A-DON.

The level of 3-A-DON significantly exceeded that of DON. The highest amounts of both toxins were observed in the control: 4 µg per g dry weight (DW) for DON, and 114 µg per g DW for 3-A-DON. The application of dsTRI5RNA decreased DON accumulation by 53–85% (Figure 1a, Appendix A). Treating the fungus with 0.96 µg of dsTRI5RNA had the strongest effect on DON accumulation, and reduced DON content by 85%; this was also the only treatment for which the effect could be determined as significant.

3-A-DON was significantly reduced in all treatments by 61–87% (Figure 1b). Interestingly, the pattern of 3-A-DON biosynthesis under various concentrations of dsTRI5RNA was the same as for DON. The application of 0.96 µg dsRNA had the strongest effect on 3-A-DON accumulation.

Treatment with dsTRI5RNA led to the downregulation of *TRI5* by 84–97% (Figure 1c). Interestingly, the highest concentration of dsTRI5RNA had the least effect on gene silencing.

We also determined that the expression of *TRI5* had a positive correlation with DON and 3-A-DON production. The Pearson correlation coefficients amounted to 0.96 and 0.98, respectively.

### 3.2. Silencing of the TRI5 Gene in the Wheat-F. culmorum Pathosystem Using VIGS by BSMV

To test whether the silencing of *TRI5* can affect *F. culmorum* aggressiveness directly on wheat plants, the barley stripe mosaic virus (BSMV) vector system was used. The γ-vector was modified to contain a 161 bp fragment of *TRI5*.

#### 3.2.1. Inoculation of Wheat Plants with the BSMV Constructs

The experimental design included four variants: a negative control (mock infection, virus-free plants), and the inoculation of wheat seedlings with either the wild type virus, the virus carrying the fragment of interest i.e. the *TRI5* sequence, or the virus carrying a fragment of the phytoene desaturase gene (*PDS*). The latter was used as a control to ensure the ability of the BSMV system to silence targeted genes. The inoculation was performed as described in Section 2.7.

The effectiveness of inoculation was quite high (98%). Plants infected with the empty virus and the one carrying the *TRI5* fragment developed yellow stripes, proving the spread of the virus within plants. Plants inoculated with the virus carrying the *PDS* sequence showed white stripes or bleached regions distinctly different in color from the other virus constructs (Figure 2), thus demonstrating effective *PDS* silencing.

#### 3.2.2. Evaluation of Wheat Spikelet Infection

Wheat heads were point-inoculated with *F. culmorum* conidia at the time of highest susceptibility, the flowering period. FHB spread was evaluated at 7, 12, 16, and 20 days post infection (dpi). Plants infected with the *TRI5* construct showed significantly fewer symptoms (Figure 3a). Pictures were taken at 20 dpi, when wheat heads were harvested for RNA extraction. It was clear that the fungus was able to penetrate the plant tissue at the point of inoculation, whereas its further growth was restricted. Some plants had black necrotic regions, others bleached ones. Both are typical symptoms of *Fusarium* spp. Plants without virus were heavily affected by FHB. Two wheat varieties with differing tolerances to *F. culmorum*, Zlata and Ivolga, were used in the experiment.

Since both genotypes showed a similar response to *F. culmorum* colonization in all variants, we combined the data to compare FHB severity. The dynamics of spikelet infection are presented in Figure 3b. The proportion of infected spikelets was reduced by 73% with the *TRI5* construct. Although infection with the wild type virus also resulted in a significant reduction in FHB symptoms by 47%, the effect in the presence of the *TRI5* construct was significantly stronger, indicating that *TRI5* silencing had an effect on virulence.

#### 3.2.3. *TRI5* Expression as Affected by Different Viral Constructs

Since, in previous experiments, problems arose in detecting *TRI5* transcripts in samples from plants harvested at 20 dpi, we decided to collect the material for real-time PCR at 14 dpi on wheat heads taken from 40 plants in total.

The levels of gene expression were similar both in the control plants and those infected by the empty virus (Figure 4). However, a significant doubling of upregulation of *TRI5* was observed in plants carrying the virus with the *TRI5* construct.

### 3.3. SIGS by Spraying of Detached Wheat Leaves with dsTRI5RNA and Its Effect on F. culmorum Virulence

dsTRI5RNA (161 bp fragment) was synthesized using the in vitro transcription kit (Section 2.6). Leaves were sprayed with dsRNA and point-inoculated with *F. culmorum* conidia as described in Section 2.10. Each plate contained two leaves. In total, six plates with leaves from two genotypes were used per treatment.

The application of 10 µg of dsTRI5RNA inhibited fungal growth, while the control plants were noticeably infected by *F. culmorum* (Figure 5a). The symptoms observed were bleached areas, with occasional small necrotic regions.

The qPCR results showed that leaves treated with dsTRI5RNA contained 5 times less fungal biomass (Figure 5b). The spraying of 10 µg of dsTRI5RNA reduced *TRI5* expression by 76% (Figure 5c).

Since no significant genotype effect was observed either with *TRI5* VIGS or SIGS, we decided to conduct the rest of the experiments only on one genotype, Zlata. dsRNA was diluted in sterile water and the following quantities were used: 15, 1.5, 0.15, 0.015 µg. To ensure that the double stranded molecule does not affect the results by itself, another control was introduced – dsRNA synthesized from a template of plasmid DNA, with no specificity to either the fungal or wheat genome (non-target dsRNA). Such dsRNA was prepared in the highest amount used in this experiment – 15 µg. Two leaves were used per plate. In total, three plates were used per treatment.

Both control variants were affected by the pathogen (Figure 6a). Leaves treated with dsTRI5RNA showed different infection patterns, but appeared less severely infected. Several leaves had necrotic areas surrounded by bleached regions spreading further on the surface.

A possible reduction in fungal biomass of 92%, 95%, and 87% was observed with 15, 1.5, and 0.015 µg dsTRI5RNA, respectively. The application of 0.15 µg did not seem to have any effect (Figure 6b). Due to strong variability, however, we could not find statistical significance for this trend.

The expression of *TRI5* did not differ significantly between both control variants. However, spraying of 15, 1.5, 0.15, and 0.015 µg of dsTRI5RNA led to *TRI5* up-regulation by almost 6, 7, 2, and 10 times, respectively (Figure 6c).

### 3.4. The Effect of SIGS with dsTRI5RNA on FHB Development on Wheat Heads

Wheat heads were treated with different concentrations of dsTRI5RNA at flowering time, followed by point inoculation with *F. culmorum* conidia. The dynamics of spikelet infection were estimated at 7, 14, and 20 dpi. The same concentrations were used as in the detached leaf assay: 15, 1.5, 0.15, and 0.015 µg. Non-target dsRNA was used as a control for non-specific gene silencing. Both controls developed strong FHB symptoms. Based on the visual evaluation, the lowest concentration of 0.015 µg was less effective than the others; the wheat heads had large bleached regions. Plants treated with dsTRI5RNA showed milder symptoms (Figure 7a).

Concentrations of 15, 1.5, and 0.15 µg of dsTRI5RNA significantly reduced the spread of *F. culmorum* within wheat heads by 47, 65, and 73%, respectively (Figure 7b). The application of 0.015 µg did not differ significantly from the controls.

At 14 dpi, wheat heads were collected from half of the plants (out of 36 plants in total) for RNA extraction and further analyses via qPCR. *F. culmorum* quantification by qPCR showed a reduction of 80 and 70%, for 15 µg and 1.5 µg of dsTRI5RNA respectively. We did not find statistical significance for this highly promising tendency, however (Figure 8a).

Simultaneously, *TRI5* up-regulation was observed following the application of 15 and 1.5 µg of dsTRI5RNA (Figure 8b, also not significant).

## 4. Discussion

Earlier findings indicate that trichothecene mycotoxins have no significant effect on *Fusarium* spp. growth in vitro. Field strains and knock-out mutants unable to produce trichothecenes showed the same growth rate as toxin-producing strains [46]. In our experiment, the application of dsTRI5RNA into liquid media did not inhibit *F. culmorum* growth either. Visually, the treated variants looked very similar to the control.

However, the silencing of house-keeping genes can affect the well-being of a pathogen. Koch et al. [19] reported the suppression of *F. graminearum* growth and changes in morphology when downregulating *CYP51* expression and, therefore, disrupting ergosterol biosynthesis. The silencing of the myosin-5 gene (*MYO5*) resulted in cell wall defects in *Fusarium asiaticum,* negatively affecting its growth, vitality, and virulence [34]. Since *F. asiaticum* is not able to form secondary siRNA, gene expression went back to normal after all exogenous siRNA was used. A regular supply of media with dsRNA could maintain *MYO5* silencing for 7 days [34].

Similar patterns were observed in this study, both for trichothecene biosynthesis and *TRI5* expression when adding dsTRI5RNA into liquid media with *F. culmorum*. The high determination coefficient (R^2^) between TRI5RNA and DON/3-A-DON production indicated the direct effect of *TRI5* silencing on trichothecene biosynthesis. Interestingly, high dsTRI5RNA concentrations seemed to be less effective than lower ones. We suspect that when an excessive quantity of dsTRI5RNA was used, some proportion was involved in other fungal-related processes.

When using viral vectors to silence *TRI5*, a few modifications had to be made to the VIGS-protocol. In earlier studies, as well as in the protocol used, it was suggested that plants be infected with a pathogen 10–14 days after the infection with the modified BSMV. However, even though *F. culmorum* is able to infect any plant organ, its main entry point is the wheat head. Therefore, we decided to carry out the infection when plants start flowering and wheat is the most sensitive. The strongest BSMV multiplication happens in the early infection stage when clear symptoms appear. The longer the virus stays in a plant organism, the more mechanisms of host defense are triggered as the infection becomes chronic. In this phase, symptoms become weaker and virus concentration decreases [21]. Without doubt, plants infected with the BSMV-TRI5 construct tolerated FHB far better. At the same time, plants carrying the empty virus were also less affected by FHB than the control. We believe that the virus itself was a serious stress-factor for plants, inducing the activation of a series of defense reactions that negatively affected *F. culmorum* spread. It is important to mention that the interaction among three biological units (plant-virus-fungus) is very complex and, therefore, hard to predict. However, a significant difference in spikelet contamination dynamics between the empty-virus variant and BSMV-TRI5 plants does support the impact of dsTRI5RNA on *F. culmorum* virulence.

Sensitive plants exhibit the rapid spread of *Fusarium* spp. from one infected spikelet. On the other hand, fungal growth can be restricted to the point of inoculation on resistant plants even under favorable conditions [3,47]. Various studies report different levels of virulence of *Fusarium* spp. to be affected by the capacity for trichothecene production. Some strains unable to produce trichothecenes are less virulent on wheat and maize [3,5,8,48]. However, it seems that the presence of *Fusarium* toxins has no influence on triticale resistance, suggesting different mechanisms of host immune suppression [49,50].

Spraying detached wheat leaves with 10 µg of dsTRI5RNA negatively influenced *F. culmorum* growth, suggesting the importance of DON and 3-A-DON in virulence. This was determined by both visual evaluation and qPCR. The analysis of mTRI5RNA levels also showed *TRI5* downregulation during dsTRI5RNA treatment. Additionally, qPCR showed a decrease in fungal biomass when spraying detached leaves with different concentrations of dsRNA. Interestingly, only one of the experiments showed significance. Unlike the visual assessment, the dilution series experiment followed by qPCR showed no significance. Additionally, *F. culmorum* quantity was no different from the controls after spraying of 0.15 µg, which was not even the lowest concentration. *Fusarium* spp. can also grow on the plant surface without actual penetration of the tissue. First, fungal hyphae grow over the surface forming a dense network, followed by the opportunistic invasion of plant tissues, for example, when the plant becomes more susceptible [1]. In this case, we do not expect to see symptoms or hyphal growth in visual assessment (without using a microscope), but qPCR would surely detect the fungus. Therefore, qPCR results without any additional visual evaluation or microscopy would not be informative enough. Since we presume that trichothecenes play a role in *F. culmorum* aggressiveness only as an additional virulence factor, its growth would not be affected by *TRI5* silencing unless the pathogen is in direct conflict with the plant immune system [1,3,4,5]. It is likely that the presence of dsTRI5RNA was the very reason why *F. culmorum* grew on the surface in this variant, until the dsTRI5RNA was gone. The non-target dsRNA seemed to have no effect on the pathogen, since the abundance of *F. culmorum* did not differ from the control.

A similar pattern was observed when spraying wheat heads with dsTRI5RNA. Visual evaluation showed that the three highest concentrations significantly suppressed fungal spread within wheat heads. The application of 0.015 µg of dsTRI5RNA was probably insufficient due to rapid degradation. The stability of dsRNA is significantly influenced by various RNAses present on non-sterile plant surfaces [51]. For example, Song et al. [34] reported that the stability of dsRNA sprayed on plants was restricted to 8 days. qPCR showed that the amount of fungal mass under 0.15 µg of dsTRI5RNA was no different from the controls or 0.015 µg, however, plants were visually less affected by FHB. Again, we hypothesize that qPCR detected the fungus growing on a head surface. Under treatment with 15 and 1.5 µg, the decrease in fungal quantity amounted to 80 and 70%, supporting the results of the visual assessment. Perhaps *F. culmorum* could not produce sufficient trichothecene and so stayed on the plant surface in order to survive without direct interaction with the host immune system. It has been reported that *Fusarium* spp. are able to grow or maintain spore vitality on various surfaces such as plant residues under unfavorable conditions, e.g., during winter [52].

We believe that the effectiveness of 15 and 1.5 µg of dsTRI5RNA could be associated with the better immune response of the crop under low trichothecene levels. Just as trichothecenes are poisonous for humans and animals, they are also toxic for plants. These toxins can inhibit protein, DNA, and RNA biosynthesis [2,53], and negatively affect cell division [54] and mitochondrial functions [55]. High DON concentration can cause the complete loss of chloroplast pigments [56]. The growth of *Arabidopsis thaliana* was significantly decreased by DON in nutrient media; its detached leaves were damaged by this compound as well [57]. We presume that in the first days of inoculation following dsTRI5RNA application, *TRI5* expression was somewhat downregulated. In this situation, *F. culmorum* could not suppress the host’s defense mechanisms and the plant was able to tolerate pathogen more effectively.

Earlier dsRNA spraying has been effectively used against *F. graminearum* and *F. asiaticum* [24,34]. It was shown that the dsRNA spread through the plant vascular system and was eventually absorbed by the fungus through the leaf tissue [24]. SIGS was also tested in insect controls [23,58,59,60]. Interestingly, not only top spraying but also root treatment was effective in the silencing of pest and pathogen genes [61].

Significantly increased levels of expression of *TRI5*, observed with VIGS by BSMV, drew our attention. The high standard deviation indicates the unstable production of mTRI5RNA, as compared to the control and the variant carrying the empty virus. A similar pattern was observed after leaf and head spraying with different concentrations of dsTRI5RNA. On detached leaves, *TRI5* upregulation was also observed, but only when spraying with the highest dsTRI5RNA amount. Interestingly, the more fungus was present, the less mTRI5RNA transcripts were detected. Moreover, such variants did not show high variation in transcript quantity, which suggests stable *TRI5* expression. In contrast, variants with a low fungal biomass or weaker FHB symptoms were characterized by high and unstable mTRI5RNA production, expressed in terms of standard deviation. If *F. culmorum* uses trichothecenes to weaken its host, it is safe to assume that the fungus produces them at an early stage of infection. Once the plant immune system has been suppressed, the fungus does not have to produce more toxins, and by the end of the growing season, most *TRI5* transcripts degrade. Since the gene expression of the treated variants is measured relative to the control, if the control contains only residues of mTRI5RNA, upregulation in the treated variants could be determined.

Even though dsTRI5RNA potentially inhibits the production of trichothecenes early on, as dsTRI5RNA is processed both by RNAi and RNAses, the expression of *TRI5* would be restored gradually. Perhaps the artificial suppression of toxin production had a stimulating effect on *TRI5* expression, and when the dsTRI5RNA quantity became insufficient for gene silencing, *F. culmorum* started intensive trichothecene secretion as a required measure to survive and propagate within plant tissues. Interestingly, several fungicides can trigger active toxin accumulation under certain conditions while disease symptoms are weak [11].

## 5. Conclusions

The application of dsTRI5RNA to liquid *F. culmorum* culture resulted in significant *TRI5* silencing and a reduction in DON and 3-A-DON secretion. VIGS and/or SIGS led to reduced infections on wheat heads or wheat leaves, respectively. The fact that the silencing of *TRI5* could not be observed in all experiments may be due to later upregulation of the gene, in an attempt by the pathogen to compensate for earlier gene downregulation. Taken together, our results indicate that by silencing of *TRI5,* it is possible to reduce trichothecene contaminations in infected plants and also the virulence of *F. culmorum*. Future efforts should be directed towards identifying the optimal amount of dsTRI5RNA used in SIGS and, as for other targets, formulations to keep dsRNA stable.

The silencing of *TRI5* can be viewed as a measure of fungistatic rather than fungicidal effect, suppressing *F. culmorum* growth in wheat at a certain time during a growing season. Therefore, dsTRI5RNA could potentially be used in mixtures with other dsRNAs aimed at *F. culmorum* house-keeping genes. Thus, our findings can contribute to novel RNAi-based control strategies against *F. culmorum*, which can be both environmentally friendly and cost efficient.

## Figures and Tables

**Figure 1 genes-13-00395-f001:**
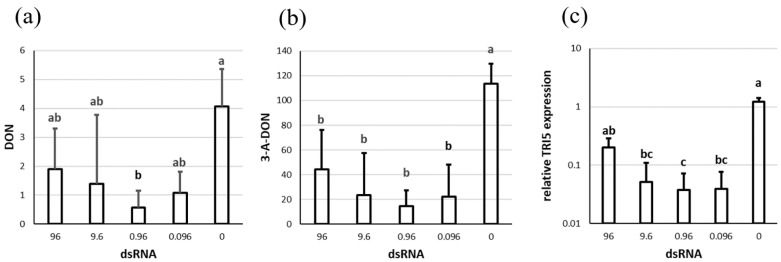
Trichotecene accumulation and *TRI5* expression after silencing with dsTRI5RNA. (**a**) DON concentration and (**b**) 3-A-DON concentration in µg per g DW. (**c**) *TRI5* expression relative to control sample and *EF1A* as a reference gene. The x-axis shows the amount of dsRNA (in µg) that was added to the cultures. The figure shows the means and standard deviation of three replicates. Letters above the columns indicate statistical significance; *p*-value ≤ 0.05.

**Figure 2 genes-13-00395-f002:**
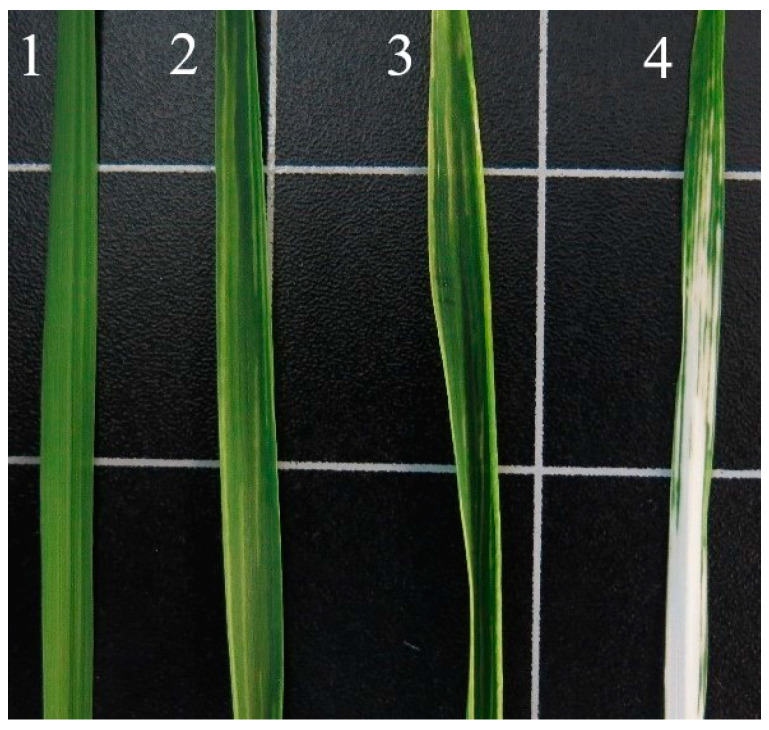
Wheat leaves infected with virus constructs. 10–12 dpi. 1—negative control, 2—empty vector, 3—BSMV-TRI5, 4—BSMV-PDS.

**Figure 3 genes-13-00395-f003:**
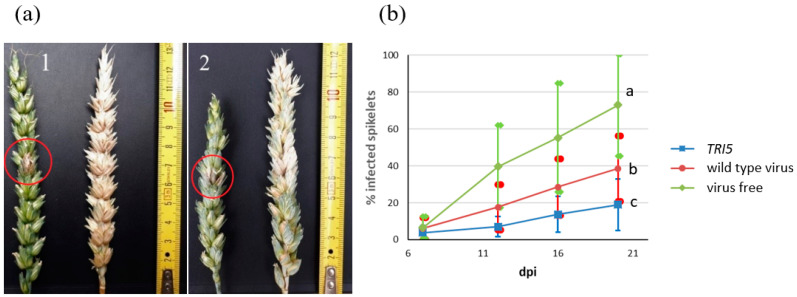
(**a**) FHB spread on wheat head at 20 dpi. 1: variety Zlata, 2: variety Ivolga. On the left: plant infected with the *TRI5* construct, on the right: negative control. The red cycle indicates the spikelet that was inoculated with *F. culmorum*. (**b**) Dynamics of spikelet infection. The graph shows the percentage of infected spikelets at 7, 12, 16, and 20 dpi, presenting the means and standard deviations of two independent experiments with at least six replicates each. Virus free plants were mock inoculated with water, wild type virus plants with BSMV RNAs α + β + γ, and TRI5 plants with BSMV RNAs α + β + γ-TRI5; for more details see Section 2.7. Letters to the right of the graphs indicate statistical significance; *p*-value ≤ 0.05.

**Figure 4 genes-13-00395-f004:**
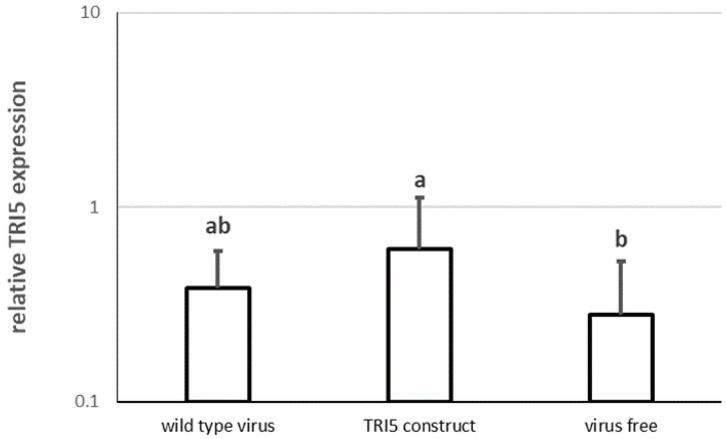
Expression of the *TRI5* gene in *F. culmorum* growing on plants infected with different viral constructs. Expression is presented relative to the control sample, using *EF1A* as a reference gene. The graph shows the means and standard deviations of seventeen plant heads for the *TRI5* construct, nine plant heads for the wild-type virus and fourteen plant heads for the virus-free variant. Virus free plants were mock inoculated with water, wild type virus plants with BSMV RNAs α + β + γ, and TRI5 plants with BSMV RNAs α + β + γ-TRI5; for more details see Section 2.7. Letters above the columns indicate statistical significance; *p*-value ≤ 0.05.

**Figure 5 genes-13-00395-f005:**
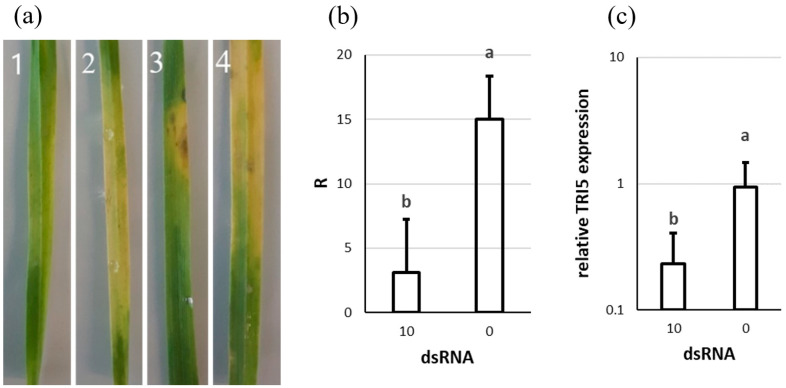
The effect of dsTRI5RNA-spraying on *F. culmorum* growth on detached wheat leaves. (**a**) Pictures of representative leaves: 1: Zlata + dsTRI5RNA, 2: Zlata + water, 3: Ivolga + dsTRI5RNA, 4: Ivolga + water. (**b**) Quantification of *F. culmorum* in the leaves based on *EF1A* relative to wheat *Act*. R calculated with formula (2). (**c**) Expression of the *TRI5* gene in *F. culmorum* growing on the leaves. Expression is relative to the control sample, using *EF1A* as a reference gene. 0: water, 10: amount in µg of dsRNA. The graph shows the means and standard deviations of at least four biological replicates, or eight leaves. Letters above the columns indicate statistical significance; *p*-value ≤ 0.05.

**Figure 6 genes-13-00395-f006:**
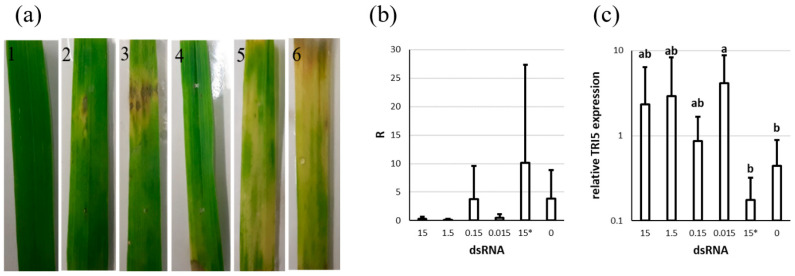
The influence of different concentrations of dsTRI5RNA on *F. culmorum* growth and *TRI5* expression on detached wheat leaves. (**a**) Representative pictures: 1: 15 µg, 2: 1.5 µg, 3: 0.15 µg, 4: 0.015 µg dsTRI5RNA, respectively; 5: 15 µg of non-target dsRNA (control); 6: water (control). (**b**) Quantification of *F. culmorum* in the leaves based on *EF1A* relative to wheat *Act*. R calculated with formula (2). (**c**) Expression of the *TRI5* gene in *F. culmorum* growing on the leaves. Expression is relative to the control sample, using *EF1A* as a reference gene. Numbers on x-axes are amounts in µg of dsRNA, * indicates non-target dsRNA. The graph shows the means and standard deviations of three biological replicates or six leaves for each treatment. Letters above the columns indicate statistical significance; *p*-value ≤ 0.05.

**Figure 7 genes-13-00395-f007:**
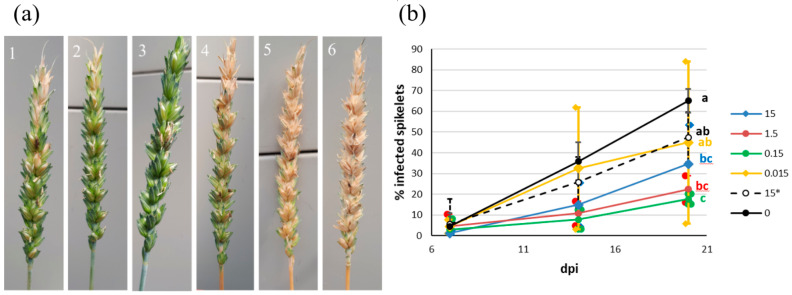
FHB after spraying with various amounts of dsTRI5RNA. (**a**) Representative pictures: 1: 15 µg, 2: 1.5 µg, 3: 0,15 µg, 4: 0,015 µg, respectively; 5: 15 µg of non-target dsRNA (control); 6: water (control). (**b**) Dynamics of spikelet infection. The graph shows the percentage of infected spikelets at 7, 14, and 20 dpi, means and standard deviations (see Section 2.10 for sample sizes). Numbers in the legend indicate the different amounts of dsTRI5RNA in µg; * indicates non-target dsRNA. Letters to the right of the graphs indicate statistical significance; *p*-value ≤ 0.05.

**Figure 8 genes-13-00395-f008:**
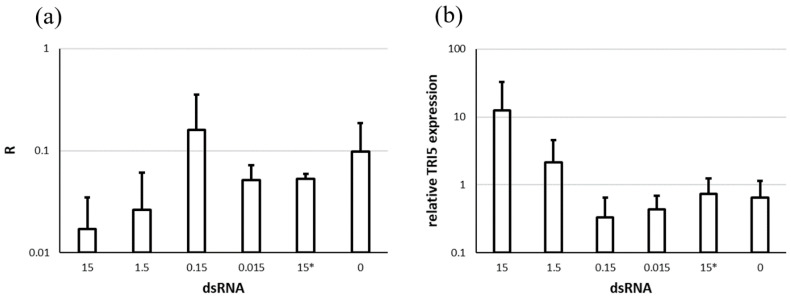
*F. culmorum* quantification and relative *TRI5* expression on wheat heads after spraying with various amounts of dsTRI5RNA. (**a**) Quantification of *F. culmorum* in the wheat heads based on *EF1A* relative to wheat *Act*. R calculated with formula (2). (**b**) Expression of the *TRI5* gene in *F. culmorum* growing in the wheat heads. Expression is relative to the control sample, using *EF1A* as a reference gene. Numbers on x-axes are amounts in µg of dsRNA, * indicates non-target dsRNA. The graph shows the means and standard deviations of at least two biological replicates or 2 heads.

**Table 1 genes-13-00395-t001:** List of primers used in this study.

Treatment		Sequence [5’–3’]	Target/[Reference]
Fc-TRI5-T7	F	TAATACGACTCACTATAGGGCTGGATTGAGCACTACAAC	*F. culmorum TRI5*
	R	TAATACGACTCACTATAGGAACGGCTGTCGTGATTTC	
Fc-TRI5-REN	F	T*TTAATTAA*GCTGGATTGAGCACTACAAC	*F. culmorum TRI5*
	R	GGTG*GCGGCCGC*AACGGCTGTCGTGATTTC	
TRI5_A	F	GTTTCATGCACGGCTACGTC	*F. culmorum TRI5*
	R	TTGGCGTCCTCTGTATCCTG	
EF1A	F	AGATTGGCGGTATTGGAACG	*F. culmorum EF1A*
	R	TTGGAAGGAGCGAAGGTAAC	
Ta54825	F	TGACCGTATGAGCAAGGAG	Wheat actin gene [39]
	R	CCAGACAACTCGCAACTTAG	
pGamma	F	TGATGATTCTTCTTCCGTTGC	BSMV γ-vector [40]
	R	TGGTTTCCAATTCAGGCATCG	

## Data Availability

Data is contained within the article or Appendix A.

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
