# Peer review of "Successful Silencing of the Mycotoxin Synthesis Gene TRI5 in Fusarium culmorum and Observation of Reduced Virulence in VIGS and SIGS Experiments"

_genes, 2022, doi:10.3390/genes13030395_

Round 1

Reviewer 1 Report

The Article, in general, is well written and competently assembled, 

The idea of the Article is good and will be of interest to readers of the journal. I recommend accepting this Article for publication.  

Author Response

First we want to acknowledge all three reviewers for their comments. Refereeing papers is a highly important task in science and it cannot be appreciated enough! Thank you.

We found that almost all comments were useful and/or justified. We did our best to improve our manuscript based on these suggestions. Below we explain the changes we made to the manuscript and we detail our responses to the reviewers.

Revisions made to the manuscript

Abstract and Keywords

As observed by Reviewer 2, our abstract represented our findings in very general terms. The results part of the abstract was written more like conclusions. To make this more concrete, we have added numbers representing our most important results.

During our revisions we also found that “spraying” is too simple a term for the spray induced gene silencing approach. We have replaced spraying with SIGS, and to complement this, we have also used the term virus induced gene silencing (VIGS) and added both terms to the Keywords.

Introduction

Reviewer 2 and Reviewer 3 both considered our introduction too long and found that it contained too much background information. To fix this we have removed all background information that did not directly relate to our experiments. Indeed, this made the introduction more concise.

Since Reviewer 2 also wanted us to mention economic impact, we have enhanced the passage describing the potential of RNAi methods (especially SIGS). The basic economic impact of our findings should, eventually, be the reduction of crop losses.

Material and Methods

Some information giving details about the experiments was transferred from results. This made some of the methods descriptions more concrete. In two instances, as requested by Reviewer 2, we added relevant citations.

Results

Due to the finding of Reviewer 3, that Methods were repeated in Results, we have carefully reviewed Results. We have deleted several redundancies and moved descriptions from Results to Methods. In some cases, we clarified the wording.

In section 3.3 we now also clearly express that measuring fungal biomass through qPCR gave no significant differences for the experiment presented in figure 6.

A similar adjustment was made for 3.4.

Discussion and conclusions

We reviewed our discussion and in several instances improved the wording. Due to the point of Reviewer 3 that the results in figure 6 are not plausible, we also adjusted the discussion part considering these findings.

We almost completely rewrote the conclusions and added thoughts on how our findings can contribute to more efficient crop protection in the future.

Responses to reviewer

To clearly distinguish our responses from the comments, we copied the “Comments and Suggestions for Authors
” from the Reviewer and represent them in standard font, whereas our responses are added in red color.

Reviewer 1

The Article, in general, is well written and competently assembled,

The idea of the Article is good and will be of interest to readers of the journal. I recommend accepting this Article for publication.

Thank you very much for your recommendation. We do appreciate it.

Reviewer 2 Report

The article Successful silencing of the mycotoxin synthesis gene TRI5 in Fusarium culmorum and observation of reduced virulence in HIGS and SIGS experiments is interesting and has scientific merits but certain points required more effort from the authors to meet the journal standards 

among these points: 1- In the abstract part there are no solid results that indicate the work carried out in the manuscript, that might cause a bad impact on the audience  2- The introduction is too long and contains too much intro that is not needed in the manuscript therefore a more concise intro will be better with a special emphasis on the economical impact of the work 3- Some experiments uses have no references that required to be adjusted 4- The magnification power in the photos of the plants in the figures are totally missed 5- The raw data for the rt-PCR should be given as supplementary 

Author Response

First we want to acknowledge all three reviewers for their comments. Refereeing papers is a highly important task in science and it cannot be appreciated enough! Thank you.

We found that almost all comments were useful and/or justified. We did our best to improve our manuscript based on these suggestions. Below we explain the changes we made to the manuscript and we detail our responses to the reviewers.

Revisions made to the manuscript

Abstract and Keywords

As observed by Reviewer 2, our abstract represented our findings in very general terms. The results part of the abstract was written more like conclusions. To make this more concrete, we have added numbers representing our most important results.

During our revisions we also found that “spraying” is too simple a term for the spray induced gene silencing approach. We have replaced spraying with SIGS, and to complement this, we have also used the term virus induced gene silencing (VIGS) and added both terms to the Keywords.

Introduction

Reviewer 2 and Reviewer 3 both considered our introduction too long and found that it contained too much background information. To fix this we have removed all background information that did not directly relate to our experiments. Indeed, this made the introduction more concise.

Since Reviewer 2 also wanted us to mention economic impact, we have enhanced the passage describing the potential of RNAi methods (especially SIGS). The basic economic impact of our findings should, eventually, be the reduction of crop losses.

Material and Methods

Some information giving details about the experiments was transferred from results. This made some of the methods descriptions more concrete. In two instances, as requested by Reviewer 2, we added relevant citations.

Results

Due to the finding of Reviewer 3, that Methods were repeated in Results, we have carefully reviewed Results. We have deleted several redundancies and moved descriptions from Results to Methods. In some cases, we clarified the wording.

In section 3.3 we now also clearly express that measuring fungal biomass through qPCR gave no significant differences for the experiment presented in figure 6.

A similar adjustment was made for 3.4.

Discussion and conclusions

We reviewed our discussion and in several instances improved the wording. Due to the point of Reviewer 3 that the results in figure 6 are not plausible, we also adjusted the discussion part considering these findings.

We almost completely rewrote the conclusions and added thoughts on how our findings can contribute to more efficient crop protection in the future.

Responses to reviewer

To clearly distinguish our responses from the comments, we copied the “Comments and Suggestions for Authors
” from the Reviewer and represent them in standard font, whereas our responses are added in red color.

Reviewer 2

The article Successful silencing of the mycotoxin synthesis gene TRI5 in Fusarium culmorum and observation of reduced virulence in HIGS and SIGS experiments is interesting and has scientific merits but certain points required more effort from the authors to meet the journal standards

among these points:

Thank you for your comments. We did our best to make the corresponding changes as can be viewed in the manuscript file. Below we comment on the more detailed points made:

  • In the abstract part there are no solid results that indicate the work carried out in the manuscript, that might cause a bad impact on the audience

Indeed, our abstract represented the results only in very general terms. We have reviewed the abstract and added concrete numbers for the most important experiments.

  • The introduction is too long and contains too much intro that is not needed in the manuscript therefore a more concise intro will be better with a special emphasis on the economical impact of the work

We have been aiming for an introduction that gives a nice overview over RNAi, both the basics and recent developments. However, this has led, as you have observed, to the inclusion of some background information that is not directly connected to our study. We have removed all extra information and considerably shortened our introduction. We also added some sentences describing that not only can RNAi be very environmentally friendly but the targeted development of new agents can also be very cost effective.

  • Some experiments uses have no references that required to be adjusted

We have added references for the SIGS procedure and for point inoculations of ears. Else we did not think additional references are necessary.

  • The magnification power in the photos of the plants in the figures are totally missed

Indeed, we are including a scale in Figure 3 (a), that is not present elsewhere. However, all our pictures are just standard photographs of leaves or heads, we did not use loupes or microscopes for magnification. So, deducing the actual sizes of the leaves and heads should be no problem for readers. Giving magnification data for standard photographs also is not very common as far as we can tell.

  • The raw data for the rt-PCR should be given as supplementary

Maybe this comment was caused by the fact that in the text we are only once referring to Table S1. But here we also put data underlying the other figures. As far as we can tell we have included all relevant data in supplementary. If we have overlooked something it would be very helpful if you could be more specific in this instance.

Reviewer 3 Report

The MS entitled “Successful silencing of the mycotoxin synthesis gene TRI5 in Fusarium culmorum and observation of reduced virulence in HIGS and SIGS experiments” is complete and well within the scope of this Journal. This study showed application of dsTRI5RNA into liquid F. culmorum culture resulted in silencing of TRI5 and the reduction of DON and 3-A-DON secretion. This study contains observations that will be of interest to the readership. However, this MS has some major issues that need to be addressed.

Major comments

  1. Introduction: The background information is more in-depth. Authors should rewrite an introduction that is clear and concise.
  2. Materials and methods are repeated many times in Results section and should rewrite the results in a clear and concise manner.
  3. There are no punctuation marks in the MS, so check carefully and put the punctuation mark where it is needed.
  4. Result: Authors stated that “A reduction in fungal biomass of 92%, 95%, and 87% was observed with of 15, 1.5 495 and 0.015 μg dsTRI5RNA, respectively. 0.15 μg did not seem to have any effect (Figure 496 6b)”. How 0.015 is effective? than 0.15 μg?. This is unbelievable. Authors should repeat the experiment.
  5. In many instances, there is no statistical analysis in the Figures.
  6. Conclusion: Authors should include future perspective of the study (should not summarize the results).

Author Response

First we want to acknowledge all three reviewers for their comments. Refereeing papers is a highly important task in science and it cannot be appreciated enough! Thank you.

We found that almost all comments were useful and/or justified. We did our best to improve our manuscript based on these suggestions. Below we explain the changes we made to the manuscript and we detail our responses to the reviewers.

Revisions made to the manuscript

Abstract and Keywords

As observed by Reviewer 2, our abstract represented our findings in very general terms. The results part of the abstract was written more like conclusions. To make this more concrete, we have added numbers representing our most important results.

During our revisions we also found that “spraying” is too simple a term for the spray induced gene silencing approach. We have replaced spraying with SIGS, and to complement this, we have also used the term virus induced gene silencing (VIGS) and added both terms to the Keywords.

Introduction

Reviewer 2 and Reviewer 3 both considered our introduction too long and found that it contained too much background information. To fix this we have removed all background information that did not directly relate to our experiments. Indeed, this made the introduction more concise.

Since Reviewer 2 also wanted us to mention economic impact, we have enhanced the passage describing the potential of RNAi methods (especially SIGS). The basic economic impact of our findings should, eventually, be the reduction of crop losses.

Material and Methods

Some information giving details about the experiments was transferred from results. This made some of the methods descriptions more concrete. In two instances, as requested by Reviewer 2, we added relevant citations.

Results

Due to the finding of Reviewer 3, that Methods were repeated in Results, we have carefully reviewed Results. We have deleted several redundancies and moved descriptions from Results to Methods. In some cases, we clarified the wording.

In section 3.3 we now also clearly express that measuring fungal biomass through qPCR gave no significant differences for the experiment presented in figure 6.

A similar adjustment was made for 3.4.

Discussion and conclusions

We reviewed our discussion and in several instances improved the wording. Due to the point of Reviewer 3 that the results in figure 6 are not plausible, we also adjusted the discussion part considering these findings.

We almost completely rewrote the conclusions and added thoughts on how our findings can contribute to more efficient crop protection in the future.

Responses to reviewer

To clearly distinguish our responses from the comments, we copied the “Comments and Suggestions for Authors
” from the Reviewer and represent them in standard font, whereas our responses are added in red color.

Reviewer 3

The MS entitled “Successful silencing of the mycotoxin synthesis gene TRI5 in Fusarium culmorum and observation of reduced virulence in HIGS and SIGS experiments” is complete and well within the scope of this Journal. This study showed application of dsTRI5RNA into liquid F. culmorum culture resulted in silencing of TRI5 and the reduction of DON and 3-A-DON secretion. This study contains observations that will be of interest to the readership. However, this MS has some major issues that need to be addressed.

Thank you for your comments. We did our best to make the corresponding changes as can be viewed in the manuscript file. Below we respond to the comments one by one:

Major comments

  1. Introduction: The background information is more in-depth. Authors should rewrite an introduction that is clear and concise.

We have been aiming for an introduction that gives a nice overview over RNAi, both the basics and recent developments. However, this has led, as you have observed, to the inclusion of some background information that is not directly connected to our study. We have removed all extra information and considerably shortened our introduction.

  1. Materials and methods are repeated many times in Results section and should rewrite the results in a clear and concise manner.

It is true that in Results we give descriptive explanations of the experiments we did. We do acknowledge your opinion that Results should be very concise. But on the other hand we think descriptions of experiments in Results improves the understanding of the experiments. This way readers do not need to refer to Materials and Methods to know what was done for every experiment. We have now carefully reviewed Results, moved some of the descriptions to Methods, and removed all wording that we found redundant.

  1. There are no punctuation marks in the MS, so check carefully and put the punctuation mark where it is needed.

We did check carefully. Some instances of missing punctuation or misspelling were found and corrected.

  1. Result: Authors stated that “A reduction in fungal biomass of 92%, 95%, and 87% was observed with of 15, 1.5 495 and 0.015 μg dsTRI5RNA, respectively. 0.15 μg did not seem to have any effect (Figure 496 6b)”. How 0.015 is effective? than 0.15 μg?. This is unbelievable. Authors should repeat the experiment.

Indeed, this result is confusing. However, the experiment was performed in biological triplicate and, therefore, we consider this to be a true observation. (Unfortunately also, on this short notice, we did not have the option to repeat the experiment.) We also describe this issue in Discussion: “Interestingly, unlike visual assessment, qPCR showed that F. culmorum quantity was no different from the controls under the spraying of 0.15 µg, which was not even the lowest concentration. Fusarium spp. can also grow on the plant surface without actual penetration of the tissue. First fungal hyphae grow over the surface forming a dense network followed by invasion of plant tissues under the right timing, for example when plant be-comes more susceptible [1]. In this case we would not be able to see symptoms or hyphal growth without any equipment but qPCR would surely detect the fungus. Therefore, qPCR results without any additional visual evaluation or microscopy would not be informative enough.”

What might have been misleading was our description in Results. Here we now point out, that although we could see the tendency that you consider implausible, there was no statistical significance to prove it. We also adapted the wording in the discussion.

  1. In many instances, there is no statistical analysis in the Figures.

To us this seems to be an incorrect observation. We did statistical analysis for all experiments. The only instances where we did not put indicators for statistical significance into the figures is when we could find no statistical significance. This was the case for the results presented in Figure 6b, and Figure 8.

    Conclusion: Authors should include future perspective of the study (should not summarize the results).

It is true that our conclusions were mostly a summary. We have strongly reworded this part, shortening the summary part and adding sentences explaining potential uses of our findings.

Round 2

Reviewer 2 Report

The article has much improved from the initial version 
The authors amended most of the required points 
It can be accepted in the present form

Reviewer 3 Report

Accept